# Joint Contrastive Learning with Infinite Possibilities

**Qi Cai**[1*]    **Yu Wang**[2*]    **Yingwei Pan**[2]    **Ting Yao**[2]    **Tao Mei**[2]

[1] University of Science and Technology of China, Hefei, China
[2] JD AI Research, Beijing, China

{cqcaiqi, feather1014, panyw.ustc, tingyao.ustc}@gmail.com, tmei@live.com

## Abstract

This paper explores useful modifications of the recent development in contrastive learning via novel probabilistic modeling. We derive a particular form of contrastive loss named Joint Contrastive Learning (JCL). JCL implicitly involves the simultaneous learning of an infinite number of query-key pairs, which poses tighter constraints when searching for invariant features. We derive an upper bound on this formulation that allows analytical solutions in an end-to-end training manner. While JCL is practically effective in numerous computer vision applications, we also theoretically unveil the certain mechanisms that govern the behavior of JCL. We demonstrate that the proposed formulation harbors an innate agency that strongly favors similarity within each instance-specific class, and therefore remains advantageous when searching for discriminative features among distinct instances. We evaluate these proposals on multiple benchmarks, demonstrating considerable improvements over existing algorithms. Code is publicly available at: https://github.com/caiqi/Joint-Contrastive-Learning.

## 1   Introduction

In recent years, supervised learning has seen tremendous progress and made great success in numerous real-world applications. By heavily relying on human annotations, supervised learning allows for convenient end-to-end training of deep neural networks, and has made human-crafted features the least popular in the machine learning community. However, the underlying feature behind the data potentially has a much richer structure than what the sparse labels or rewards describe, while label acquisition is also time-consuming and economically expensive. In contrast, unsupervised learning uses no manually labeled annotations, and aims to characterize the underlying feature distribution completely depending on the data itself. This overcomes several disadvantages that the supervised learning encounters, including overfitting of the specific tasks-led features that cannot be readily transferred to other objectives. Unsupervised learning therefore is an important stepping stone towards more robust and generic representation learning.

Contrastive learning is at the core of several advances in unsupervised learning. The use of contrastive loss dates back to [17]. In brief, the loss function in [17] runs over pairs of samples, returning low values for similar pairs and high values for dissimilar pairs, which encourages invariant features on the low dimensional manifold. The seminal work Noise Contrastive Estimation (NCE) [16] then builds up the foundation of the contemporary contrastive learning, as NCE provides rigorous theoretical justification by posing the contrastive learning problem into the "two-class" problem. InfoNCE [36] roots in the principles of NCE and links the contrastive formulation with mutual information.

However, most existing contrastive learning methods only consider *independently* penalizing the incompatibility of each single positive query-key pair at a time. This does not fully leverage the assumption that all augmentations corresponding to a specific image are statistically dependent on

each other, and are simultaneously similar to the query. In order to take this advantage of shared similarity across augmentations, we derive a particular form of loss for contrastive learning named Joint Contrastive Learning (JCL). Our launching point is to introduce dependencies among different query-key pairs, so that similarity consistency is encouraged within each instance-specific class. Specifically, our contributions include:

- We consider simultaneously penalizing multiple positive query-key pairs in regard of their "in-pair" dissimilarity. However, carrying multiple query-key pairs in a mini-batch is beyond the practical computational budget. To mitigate this issue, we push the limit and take the number of pairs to infinity. This novel formulation inherently absorbs the impact of large number of positive pairs via a principled probabilistic modeling. We could therefore approach an analytic form of loss that allows for end-to-end training of the deep network.

- We also theoretically unveil plenty of interesting interpretations behind the loss. Empirical evidences are presented that strongly echo these hypotheses.

- Empirical results show that JCL is advantageous when searching for discriminative features and JCL demonstrates considerable boosts over existing algorithms on various benchmarks.

## 2   Related Work

**Self-Supervised Learning**. Self-supervised learning is one of the mainstream techniques under the umbrella of unsupervised learning. Self-supervised learning, as its name implies, relies only on the data itself for some form of supervision. For example, one important direction of self-supervised learning focuses on tailoring algorithms for specific pretext tasks. These pretext tasks usually leave out a portion of information from the specific training data and attempt to predict the missing information from the remaining part of the training data itself. Successful representatives along this path include: relative patch prediction [6, 12, 15, 35], rotation prediction [14], inpainting [37], image colorization [11, 24, 27, 28, 46, 47], etc. More recently, numerous self-supervised learning approaches capitalizing on contrastive learning techniques start to emerge. These algorithms demonstrate strong advantages in learning invariant features: [2, 8, 18, 21, 23, 29, 34, 36, 40, 43–45, 48]. The central spirit of these approaches aims to maximize the mutual information of latent representations among different views of the images. Different approaches consider different strategies for constructing distinct views. Take for instance, in CMC [40], RGB images are converted to *Lab* color space and each channel represents a different view of the original image. In the meanwhile, different approaches also design different policies for effectively generating negative pairs, e.g., the techniques used in [8, 18].

**Semantic Data Augmentation**. Data augmentation has been extensively explored in the context of feature generalization and overfitting reduction for effective deep network training. Recent works [1, 4, 25, 38] show that semantic data augmentation is able to effectively preserve the class identity. Among these work, one observation is that variances in feature space along some certain directions essentially correspond to implementing semantic data augmentations in the ambient space [3, 33]. In [41], interpolation in the embedding space is shown effective in achieving semantic data augmentation. [42] estimates the category-wise distribution of deep features and the augmented features are drawn from the estimated distribution.

**Comparison to Existing Works**. The proposed JCL benefits from an infinite number of positive pairs constructed for each query. ISDA [42] also involves the implicit usage of infinite number of augmentations shown to be advantageous. However, both our bounding technique and the motivation fundamentally differ from ISDA. JCL aims to develop an efficient self-supervised learning algorithm in the context of contrastive learning, where no category annotation is available. In contrast, ISDA is completely a supervised algorithm. There is also a concurrent work CMC [40] that involves optimization over multiple positive pairs. However, JCL is distinct from CMC in many aspects. In comparison, we derive a rigorous bound on the loss function that enables practical implementation of backpropagation for JCL, where the number of positive pairs is pushed to the infinity. In addition, our motivation closely follows a statistical perspective in a principled way, where positive pairs are statistically dependent. We also justify the legitimacy of our proposed formulation analytically by unveiling certain mechanisms that govern the behavior of JCL. All these ingredients are absent in CMC and significantly distinguish JCL from CMC.

# 3 Method

In this section, we explore and develop the theoretical derivation of our algorithm JCL. We also characterize how the loss function behaves in a way that favors feature generalization. The empirical evidence corroborates the relevant hypotheses stemming from our theoretical analyses.

## 3.1 Preliminaries

Contrastive learning and its recent variants aim to learn an embedding by separating samples from different distributions via a contrastive loss $\mathcal{L}$. Assuming we have *query* vectors $\boldsymbol{q} \in \mathcal{R}^d$, and *key* vectors $\boldsymbol{k} \in \mathcal{R}^d$, where $d$ is the dimension of the embedding space. The objective $\mathcal{L}$ is a function that aims to reflect incompatibility of each $(\boldsymbol{q}, \boldsymbol{k})$ pair. In this regard, the key vector set $\mathcal{K}$ is constructed as a composition of positive and negative keys, i.e., $\mathcal{K} = \mathcal{K}^+ \cup \mathcal{K}^-$, where the set $\mathcal{K}^+$ comprises of positive keys $\boldsymbol{k}_i^+$ coming from the same distribution as the specific $\boldsymbol{q}_i, (i = 1, 2, ...N)$, whereas $\mathcal{K}^-$ represents the set of negative samples $\boldsymbol{k}_i^-$ from an alternative noise distribution. A desirable $\mathcal{L}$ usually returns low values when a query $\boldsymbol{q}_i$ is similar to its positive key $\boldsymbol{k}_i^+$ while it remains distinct to negative keys $\boldsymbol{k}_i^-$ in the meanwhile.

The theoretical foundation of Noise Contrastive Learning (NCE), where negative samples are viewed as noises with regard to each query, is firstly established in [16]. In [16], the learning problem becomes a "two-class" task, where the goal is to distinguish true samples out of the empirical distribution from the noise distribution. Inspired by [16], a prevailing form of $\mathcal{L}$ is presented in InfoNCE [36] based on a softmax formulation:

$$\mathcal{L} = -\frac{1}{N} \sum_i^N \log \frac{\exp(\boldsymbol{q}_i^T \boldsymbol{k}_i^+ / \tau)}{\exp(\boldsymbol{q}_i^T \boldsymbol{k}_i^+ / \tau) + \sum_{j=1}^K \exp(\boldsymbol{q}_i^T \boldsymbol{k}_{i,j}^- / \tau)}, \qquad (1)$$

where $\boldsymbol{q}_i$ is the $i^{th}$ query in the dataset, $\boldsymbol{k}_i^+$ is the positive key corresponding to $\boldsymbol{q}_i$, $\boldsymbol{k}_{i,j}^-$ is the $j^{th}$ negative key of $\boldsymbol{q}_i$. The motivation behind Eq.(1) is straightforward: training a network with parameters that could correctly distinguish positive samples from the $K$ negative samples, i.e., from the *noise* set $\mathcal{K}^- = \{\boldsymbol{k}_{i,1}^-, \boldsymbol{k}_{i,2}^- ... \boldsymbol{k}_{i,K}^-\}$. $\tau$ is the temperature hyperparameter following [8, 18].

Orthogonal to the design of the formulation $\mathcal{L}$ itself though, one of the remaining challenges is to construct $\mathcal{K}^+$ and $\mathcal{K}^-$ efficiently in an unsupervised way. Since no annotation is available in an unsupervised learning setting, one common practice is to generate independent augmented views from each single training sample, e.g., an image $\boldsymbol{x}_i$, and consider each random pairing of these augmentations as a valid positive $(\boldsymbol{q}_i, \boldsymbol{k}_i^+)$ pair in Eq.(1). In the meanwhile, augmented views of other samples $\boldsymbol{x}_j, j \neq i$ are seen as the negative keys $\boldsymbol{k}_i^-$ that form the noise distribution against the query $\boldsymbol{q}_i$. Under this construction, each image essentially defines an individual class, and each image's distinct augmentations form the corresponding instance-specific distribution. Take for instance, SimCLR [8] uses distinct images in current mini-batch as negative keys. MoCo [18] proposes the use of a queue $\mathcal{Q}$ in order to track negative samples from neighboring mini-batches. During training, each mini-batch is subsequently enqueued into $\mathcal{Q}$ while the oldest batch of samples in $\mathcal{Q}$ are dequeued. In this way, all the currently queuing samples serve as negative keys and effectively decouples the correlation between mini-batch size and the number of negative keys. Correspondingly, MoCo exclusively enjoys an extremely large number of negative samples that best approaches the theoretical bound justified in [16]. This queuing trick also allows for feasible training on a typical 8-GPU machine and achieves state-of-the-art learning performances. We therefore adopt the MoCo's approach of constructing negative keys in this paper, owing to its effectiveness and ease of implementation.

## 3.2 Joint Contrastive Learning

Conventional formulation in Eq.(1) independently penalizes the incompatibility within each $(\boldsymbol{q}_i, \boldsymbol{k}_i^+)$ pair at a time. We instead derive a particular form of the contrastive loss where multiple positive keys are simultaneously involved with regard to $\boldsymbol{q}_i$. The goal of this modification is to force various positive keys to build up stronger dependencies via the bond with the same $\boldsymbol{q}_i$. The new objective poses a tighter constraint on instance-specific features, and tends to encourage the consistent representations within each instance-specific class during the search for invariant features.

In our framework, every query $\boldsymbol{q}_i$ now needs to return a low loss value when simultaneously paired with multiple positive keys $\boldsymbol{k}_{i,m}^+$ of its own, where subscript $m$ indicates the $m^{th}$ positive key paired

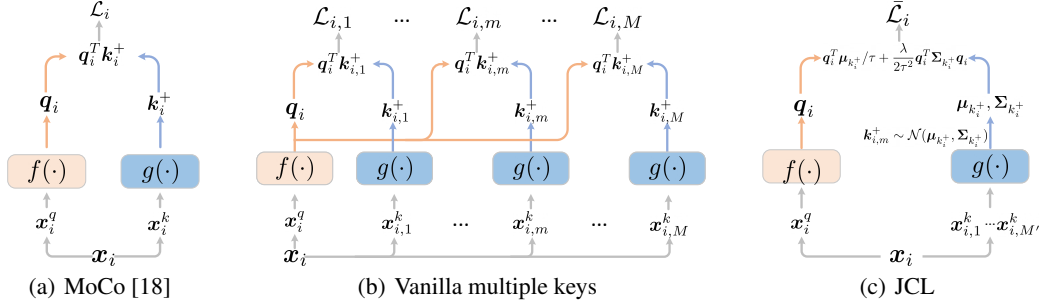

(a) MoCo [18]        (b) Vanilla multiple keys        (c) JCL

Figure 1: Conceptual comparisons of three contrastive loss mechanisms. (a) In MoCo, a single positive key $k_i^+$ is paired with a query $q_i$. (b) A vanilla extension which generates multiple keys and averages losses of all $(q_i, k_{i,m}^+)$ pairs. (c) JCL implicitly pushes the number of $k_{i,m}^+$ to infinity and minimizes the upper bound of loss expectation.

with $q_i$. Specifically, we define the loss of each pair $(q_i, k_{i,m}^+)$ as:

$$\mathcal{L}_{i,m} = -\log \frac{\exp(q_i^T k_{i,m}^+/\tau)}{\exp(q_i^T k_{i,m}^+/\tau) + \sum_{j=1}^K \exp(q_i^T k_{i,j}^-/\tau)}. \tag{2}$$

Our objective is to penalize the averaged sum of $\mathcal{L}_{i,m}$:

$$\mathcal{L}_i^M = \frac{1}{M} \sum_{m=1}^M \mathcal{L}_{i,m} \tag{3}$$

with regard to each specific query $q_i$. This procedure is illustrated in Fig.(1(b)): a specific training sample $x_i$ is firstly augmented in the ambient space into respectively: the query image $x_i^q$, and the positive key images $x_i^{k,1}, x_i^{k,2}, ... x_i^{k,m}$. Each of the query image $x_i^q$ is subsequently mapped into embedding $q_i$ via the query encoder $f(\cdot)$, while each positive key image $x_i^{k,m}$ is mapped into embedding $k_{i,m}^+$ via key encoder $g(\cdot)$. Both functions $f$ and $g$ are implemented using deep neural networks, of which the network parameters are learned during training. For comparison, Fig.(1(a)) shows the schemes of MoCo where only a single positive key is involved.

A vanilla implementation of $\mathcal{L}_i^M$ would have required the instance $x_i$ to be firstly augmented $M+1$ times ($M$ for positive keys and 1 extra for the query itself), and then to backpropagate the loss Eq.(3) via all the branches in Fig.(1(b)). Unfortunately, this is not computational applicable, as carrying all $(M+1) \times N$ pairs in a mini-batch would quickly drain the GPU memory when $M$ is even moderately small. In order to circumvent this issue, we take an infinity limit on the number $M$, where the effect of $M$ is hopefully absorbed in a probabilistic way. Capitalizing on this application of infinity limit, the statistics of the data become sufficient to reach the same goal of multiple pairing. Mathematically, as $M$ goes to infinity, $\mathcal{L}_i^M$ becomes the estimate of:

$$\mathcal{L}_i^\infty = \lim_{M \to \infty} \frac{1}{M} \sum_{m=1}^M \mathcal{L}_{i,m} = -\mathbb{E}_{k_i^+ \sim p(k_i^+)} \log \frac{\exp(q_i^T k_i^+/\tau)}{\exp(q_i^T k_i^+/\tau) + \sum_{j=1}^K \exp(q_i^T k_{i,j}^-/\tau)}. \tag{4}$$

The analytic form of Eq.(4) itself is intractable, but Eq.(4) has a rigorous closed form of upper bound, which can be derived as:

$$-\mathbb{E}_{k_i^+} \log \frac{\exp(q_i^T k_i^+/\tau)}{\exp(q_i^T k_i^+/\tau) + \sum_{j=1}^K \exp(q_i^T k_{i,j}^-/\tau)} \tag{5}$$

$$= \mathbb{E}_{k_i^+} \left[ \log \left[ \exp(q_i^T k_i^+/\tau) + \sum_{j=1}^K \exp(q_i^T k_{i,j}^-/\tau) \right] \right] - \mathbb{E}_{k_i^+} \left[ (q_i^T k_i^+/\tau) \right] \tag{6}$$

$$\leq \log \left[ \mathbb{E}_{k_i^+} \left[ \exp(q_i^T k_i^+/\tau) + \sum_{j=1}^K \exp(q_i^T k_{i,j}^-/\tau) \right] \right] - q_i^T \mathbb{E}_{k_i^+} \left[ k_i^+/\tau \right], \tag{7}$$

where Eq.(7) upperbounds $\mathcal{L}_i^\infty$. The inequality Eq.(7) emerges from the application of Jensen inequality on concave functions, i.e., $\mathbb{E}_x \log(X) \leq \log \mathbb{E}_x[X]$. This application of Jensen inequality does not interfere with the effectiveness of our algorithm and rather buys us desired optimization advantages. We analyze this part in detail in section 3.3.

**Algorithm 1** Joint Contrastive Learning

1: **Input:** batch size $N$, positive key number $M'$, queue $\mathcal{Q}$, query encoder $f(\cdot)$, key encoder $g(\cdot)$,
2: **for** sampled mini-batch $\{\mathbf{x}_i\}_{i=1}^N$ **do**
3:      **for** *each* sample $\mathbf{x}_i$ **do**
4:          randomly augment $\mathbf{x}_i$ for $M'+1$ times: $\{\boldsymbol{x}_i^q, \boldsymbol{x}_{i,1}^k, \boldsymbol{x}_{i,2}^k, ..., \boldsymbol{x}_{i,M'}^k\}$
5:          compute query representation: $\boldsymbol{q}_i = f(\boldsymbol{x}_i^q)$
6:          compute key representations: $\boldsymbol{k}_{i,m}^+ = g(\boldsymbol{x}_{i,m}^k), ..., \boldsymbol{k}_{i,M'}^+ = g(\boldsymbol{x}_{i,M'}^k)$
7:          compute average values of keys: $\boldsymbol{\mu}_{k_i^+} = \frac{1}{M'}\sum_{m=1}^{M'} \boldsymbol{k}_{i,m}^+$
8:          compute zero centered keys: $\tilde{\boldsymbol{k}}_{i,m}^+ = \boldsymbol{k}_{i,m}^+ - \boldsymbol{\mu}_{k_i^+}, m = 1, 2, ..., M'$
9:          compute covariance matrix: $\boldsymbol{\Sigma}_{k_i^+} = \frac{1}{M'}[\tilde{\boldsymbol{k}}_{i,1}^+; ...; \tilde{\boldsymbol{k}}_{i,M'}^+]^T[\tilde{\boldsymbol{k}}_{i,1}^+; ...; \tilde{\boldsymbol{k}}_{i,M'}^+]$
10:          compute loss $\bar{\mathcal{L}}_i$ based on Eq.(8)
11:      **end for**
12:      compute loss $\bar{\mathcal{L}}$ in Eq.(9) and update $f(\cdot)$, $g(\cdot)$ based on $\bar{\mathcal{L}}$
13:      enqueue $\{\boldsymbol{\mu}_{k_i^+}\}_{i=1}^N$ and dequeue oldest keys in $\mathcal{Q}$
14: **end for**
15: **return** $f(\cdot)$

To facilitate our formulation, we need some further assumptions on the generative process of $\boldsymbol{k}_i^+$ in the feature space $\mathcal{R}^d$. Specifically, we assume the variable $\boldsymbol{k}_i^+$ follows a Gaussian distribution $\boldsymbol{k}_i^+ \sim \mathcal{N}(\boldsymbol{\mu}_{k_i^+}, \boldsymbol{\Sigma}_{k_i^+})$, where $\boldsymbol{\mu}_{k_i^+}$ and $\boldsymbol{\Sigma}_{k_i^+}$ are respectively the mean and the covariance matrix of the positive keys for $\boldsymbol{q}_i$. This Gaussian assumption explicitly poses statistical dependencies among all the $\boldsymbol{k}_i^+$s, and makes the learning process appealing to consistency between positive keys. We argue that this assumption is legitimate as positive keys more or less share similarities in the embedding space around some mean value as they all mirror the nature of the query to some extent. Also there are certainly some reasonable variances expected in each feature dimension that reflects the semantic difference in the ambient space [3, 33]. In brief, we randomly augment each $\boldsymbol{x}_i$ in the ambient space (e.g., pixel values for images) for $M'$ times ($M'$ is relatively small) and compute the covariance matrix $\boldsymbol{\Sigma}_{k_i^+}$ on the fly. Since the statistics are more informative in the later of the training/less informative in the beginning of the training, we scale the influence of $\boldsymbol{\Sigma}_{k_i^+}$ by multiplying it with a scalar $\lambda$. This tuning of $\lambda$ hopefully stabilizes the training. Under this Gaussian assumption, Eq.(7) eventually reduces to (see supplementary material for more detailed derivations):

$$\bar{\mathcal{L}}_i = \log\left[\exp(\boldsymbol{q}_i^T\boldsymbol{\mu}_{k_i^+}/\tau + \frac{\lambda}{2\tau^2}\boldsymbol{q}_i^T\boldsymbol{\Sigma}_{k_i^+}\boldsymbol{q}_i) + \sum_{j=1}^K \exp(\boldsymbol{q}_i^T\boldsymbol{k}_{i,j}^-/\tau)\right] - \boldsymbol{q}_i^T\boldsymbol{\mu}_{k_i^+}/\tau. \quad (8)$$

The overall loss function with regard to each mini-batch ($N$ is the batch size) therefore boils down to the closed form whose gradients can be analytically solved for:

$$\bar{\mathcal{L}} = \frac{1}{N}\sum_{i=1}^N \bar{\mathcal{L}}_i. \quad (9)$$

Algorithm 1 summarizes the algorithmic flow of the JCL procedure. It is important to note that, the computational cost when using $M'$ number of positive keys to compute the sufficient statistics, is fundamentally different from backpropagating losses of $(M'+1) \times N$ pairs (which vanilla formulation shown as in Fig.(1(b)) would have done) from the perspective of memory schedules and cost (see more detailed comparisons in supplementary material). For comparison, we illustrate the actual JCL computation in Fig.(1(c)).

### 3.3 Analysis

We emphasize that the introduction of Jensen inequality in Eq.(7) actually unveils a number of interesting interpretations behind the loss. Firstly, by virtue of the Jensen inequality, the equality in Eq.(7) holds if and only if the variable $\boldsymbol{k}_i^+$ is a *constant*, i.e., when all the positive keys $\boldsymbol{k}_{i,m}^+$ of $\boldsymbol{q}_i$ produce identical embedding $\boldsymbol{k}_{i,m}^+$. This translates into a desirable incentive: in order to close the gap between Eq.(6) and Eq.(7) so that the loss is decreased, the training process mostly favors invariant representation across different positive keys, i.e., very similar $\boldsymbol{k}_{i,m}^+$s given different augmentations.

Also, the loss reserves a strong incentive to push queries away from noisy negative samples, as the loss is monotonously decreasing as $\sum_{j=1}^{K} \exp(\boldsymbol{q}_i^T \boldsymbol{k}_{i,j}^-)$ reduces. Most importantly, after some basic manipulation, it is easy to show that $\tilde{\mathcal{L}}_i$ is also monotonously decreasing into the direction where $\boldsymbol{q}_i^T \boldsymbol{\mu}_{k_i^+}$ increases, i.e., when $\boldsymbol{q}_i$ and $\boldsymbol{\mu}_{k_i^+}$ closely resembles each other.

We argue that conventional contrastive loss does not enjoy similar merits. Although as the training proceeds with more epochs, the $\boldsymbol{q}_i$ might be randomly paired with a numerous distinct $\boldsymbol{k}_i^+$, the loss Eq.(1) simply goes downhill as long as each $\boldsymbol{q}_i$ aligns independently with each positive key $\boldsymbol{k}_i^+$ at a time. This likely confuses the learning procedure and sabotage the effectiveness in finding a unified direction for all positive keys.

## 4    Experiments

In this section, we empirically evaluate and analyze the hypotheses that directly emanated from the design of JCL. One important purpose of unsupervised learning is to pre-train features that can be transferred to downstream tasks. Correspondingly, we demonstrate that in numerous downstream tasks related to classification, detection and segmentation, JCL exhibits strong advantages and surpasses the state-of-the-art approaches. Specifically, we perform the pre-training on ImageNet1K [10] dataset that contains 1.2M images evenly distributed across 1,000 classes. Following the protocols in [8, 18], we verify the effectiveness of JCL pre-trained features via the following evaluations: **1)** Linear classification accuracy on ImageNet1K. **2)** Generalization capability of features when transferred to alternative downstream tasks, including object detection [5, 39], instance segmentation [19] and keypoint detection [19] on the MS COCO [31] dataset. **3)** Ablation studies that reveal the effectiveness of each component in our losses. **4)** Statistical analysis on features that validates our hypothesis and proposals in the previous sections. For more detailed experimental settings, please refer to the supplementary material.

### 4.1    Pre-Training Setups

We adopt ResNet-50 [20] as the backbone network for training JCL on the ImageNet1K dataset. For the hyper-parameters, we use positive key number $M' = 5$, softmax temperature $\tau = 0.2$ and $\lambda = 4.0$ in Eq.(8) (see definitions in section 3.2). We also investigate the impact of these hyper-parameters tuning in section 4.4. Other network and parameter settings strictly follow the implementations in MoCo v2 [9] for fair apple to apple comparisons. We attach a two-layer MLP (Multiple Layer Perceptrons) on top of the global pooling layer of ResNet-50 for generating the final embeddings. The dimension of this embedding is $d = 128$ across all experiments. The batch size is set to $N = 512$ that enables applicable implementations on an 8-GPU machine. We train JCL for 200 epochs with an initial learning rate of $lr = 0.06$ and $lr$ is gradually annealed following a cosine decay schedule [32].

### 4.2    Linear Classification on ImageNet1K

**Setup.** In this section, we follow [8, 18] and train a linear classifier on frozen features extracted from ImageNet1K. Specifically, we initialize the layers before the global poolings of ResNet-50 with the parameter values obtained from our JCL pre-trained model, and then append a fully connected layer on top of the resultant ResNet-50 backbone. During training, the parameters of the backbone network are frozen, while only the last fully connected layer is updated via backpropagation. The batch size is set as $N = 256$ and the learning rate $lr = 30$ at this stage. In this way, we essentially train a linear classifier on frozen features. The classifier is trained for 100 epochs, while the learning rate $lr$ is decayed by 0.1 at the $60^{th}$ and the $80^{th}$ epoch respectively.

**Results.** Table 1 reports the top-1 accuracy and top-5 accuracy in comparison with the state-of-the-art methods. Existing works differ considerably in model size and the training epochs, which could significantly influence the performance (up to 8% in [18]). We therefore only consider comparisons to the published models of similar model size and training epochs. As Table 1 shows, JCL performs the best among all the presented approaches. Particularly, JCL outperforms all *non-contrastive learning based* counterparts by a large margin, which demonstrates evident advantages brought by the idea of contrastive learning itself. The introduction of positive and negative pairs more effectively recovers each instance-specific distributions. Most notably, JCL remains competitive and surpasses all its *contrastive learning based* rivals, e.g., MoCo and MoCo v2. This superiority of JCL over its MoCo baselines clearly verifies the advantage of our proposal Eq.(8) via the joint learning process across numerous positive pairs.

Table 1: Accuracy of linear classification model on ImageNet1K. [†] represents results from [26], which reports better performances than the original papers. [‡] means accuracies of models trained 200 epochs for fair comparisons. [§] denotes results of our re-implemented linear classifier based on pre-trained model from https://github.com/facebookresearch/moco for extracting features.

| Method | architecture | params (M) | accuracy@top1 | accuracy@top5 |
|---|---|---|---|---|
| RelativePosition [12] | ResNet-50(2x) | 94 | $51.4^{\dagger}$ | $74.0^{\dagger}$ |
| Jigsaw [35] | ResNet-50(2x) | 94 | $44.6^{\dagger}$ | $68.0^{\dagger}$ |
| Rotation [14] | RevNet(4x) | 86 | $55.4^{\dagger}$ | $77.9^{\dagger}$ |
| Colorization [46] | ResNet-101 | 28 | 39.6 | / |
| DeepCluster [7] | VGG | 15 | 48.4 | / |
| BigBiGAN [13] | RevNet(4x) | 86 | 61.3 | 81.9 |
| *methods based on contrastive learning follow:* | | | | |
| InstDisc [43] | ResNet-50 | 24 | 54.0 | / |
| LocalAgg [48] | ResNet-50 | 24 | 60.2 | / |
| CPC v1 [36] | ResNet-101 | 28 | 48.7 | 73.6 |
| CPC v2 [21] | ResNet-50 | 24 | 63.8 | 85.3 |
| CMC [40] | ResNet-50 | 47 | 64.0 | 85.5 |
| SimCLR [8] | ResNet-50 | 24 | $66.6^{\ddagger}$ | / |
| MoCo [18] | ResNet-50 | 24 | $60.6^{\ddagger}$ ($60.6^{\S}$) | $83.1^{\S}$ |
| MoCo v2 [9] | ResNet-50 | 24 | $67.5^{\ddagger}$ ($67.6^{\S}$) | $88.0^{\S}$ |
| JCL | ResNet-50 | 24 | **68.7** | **89.0** |

Table 2: Performance comparisons on downstream tasks: object detection [39](left), instance segmentation [19](middle) and keypoint detection [19](right). All models are trained with $1\times$ schedule.

| model | *Faster R-CNN + R-50* | | | *Mask R-CNN + R-50* | | | *Keypoint R-CNN + R-50* | | |
|---|---|---|---|---|---|---|---|---|---|
| | $AP^{bb}$ | $AP^{bb}_{50}$ | $AP^{bb}_{75}$ | $AP^{mk}$ | $AP^{mk}_{50}$ | $AP^{mk}_{75}$ | $AP^{kp}$ | $AP^{kp}_{50}$ | $AP^{kp}_{75}$ |
| random | 30.1 | 48.6 | 31.9 | 28.5 | 46.8 | 30.4 | 63.5 | 85.3 | 69.3 |
| supervised | 38.2 | 59.1 | 41.5 | 35.4 | 56.5 | 38.1 | 65.4 | 87.0 | 71.0 |
| MoCo [18] | 37.1 | 57.4 | 40.2 | 35.1 | 55.9 | 37.7 | 65.6 | 87.1 | 71.3 |
| MoCo v2 [9] | 37.6 | 57.9 | 40.8 | 35.3 | 55.9 | 37.9 | 66.0 | 87.2 | 71.4 |
| JCL | 38.1 | 58.3 | 41.3 | 35.6 | 56.2 | 38.3 | 66.2 | 87.2 | 72.3 |

## 4.3 More Downstream Tasks

In this section, we evaluate JCL on a variety of more downstream tasks, i.e., object detection, instance segmentation and keypoint detection. The comparisons presented here cover a wide range of computer vision tasks from box-level to pixel-level, as we aim to challenge JCL from all dimensions.

**Setup.** For object detection, we adopt Faster R-CNN [39] with FPN [30] as the base detector. Following [18], we leave the BN trained and add batch normalization on FPN layers. The size of the shorter side of each image is sampled from the range [640, 800] during training and is fixed as 800 at inference time, while the longer side of the image always keeps proportional to the shorter side. The training is performed on a 4-GPU machine and each GPU carries 4 images at a time. This implementation is equivalent to batch size $N = 16$. We train all models for 90k iterations, which is commonly referred to as the $1\times$ schedule in [18]. For the instance segmentation and keypoint detection tasks, we adopt the same settings as Faster R-CNN [39] has used. We report the standard COCO metrics including AP (averaged over [0.5:0.95:0.05] $IoU$s), $AP_{50}$($IoU$=0.5) and $AP_{75}$($IoU$=0.75).

**Results.** Table 2 shows the results for three downstream tasks on MS COCO. From observation, both supervised pre-trained models (**supervised**) and unsupervised pre-trained backbones (**MoCo**, **MoCo v2**, **JCL**) exhibit a significant performance boost against the randomly initialized models (**random**). Our proposed JCL demonstrates clear superiority over the best competitor MoCo v2. When taking a closer inspection, JCL becomes particularly advantageous when a higher $IoU$ threshold criterion is used for object detection. This might attribute to a more precise sampling of positive pairs, under which JCL is able to promote a more accurate positive pairing and joint training. Notably, JCL even successfully surpasses its supervised counterparts in terms of $AP^{mk}$, whereas MoCo v2 remains inferior to the supervised pre-training approaches.

In brief, JCL has presented robust performance gain over existing methods across numerous important benchmark tasks. In the following section, we further investigate the impact of hyperparameters and provide validations that closely corroborate our hypothesis pertaining to the design of JCL.

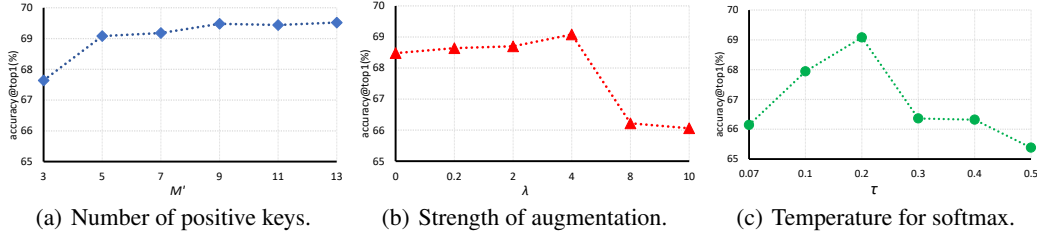

| (a) Number of positive keys. | (b) Strength of augmentation. | (c) Temperature for softmax. |

Figure 2: Performance comparisons with different hyperparameters.

## 4.4 Ablation Studies

In this section, we perform extensive ablation experiments to inspect the role of each component present in the JCL loss. Specifically, we test JCL on linear classification in ImageNet100 deployed on ResNet-18. For detailed experiment settings, please see supplementary material.

**1)** $M'$: We vary the number of positive keys used for the estimate of $\boldsymbol{\mu}_{k_i^+}$ and $\boldsymbol{\Sigma}_{k_i^+}$. By definition, larger $M'$ necessarily corresponds to a better approximation of the required statistics, although at the expense of computational complexity. From Fig.(2(a)), we observe that JCL performs reasonably well when $M'$ is in the range of [5,11], which allows for applicable GPU implementation. **2)** $\lambda$: $\lambda$ essentially controls the strength of augmentation diversity in the feature space. Larger $\lambda$ tends to inject more diverse features into the effect of positive pairing, but risks confusions with other instance distributions. Here, we vary $\lambda$ in the range of [0.0, 10.0]. Notice that in the case when $\lambda$ is marginally small, the effect of scaled covariance matrix is diminished and therefore fails to introduce the feature variance among distinct positive samples of the same query. However, an extremely large $\lambda$ overstates the effect of diversity that rather confuses the positive sample distribution with the negative samples. As the introduced variance $\boldsymbol{\Sigma}_{k_i^+}$ starts to ***dominate*** the positive mean $\boldsymbol{\mu}_{k_i^+}$ value, i.e., when the $\lambda$ is large enough to distort the magnitude scale of positive keys, the impact of negative keys in Eq.(8) is diluted. Consequently, the distribution of the $\boldsymbol{k}_{i,m}^+$ and $\boldsymbol{k}_{i,j}^-$ would also significantly be distorted. When $\lambda$ grows to infinity, the effect of negative keys completely vanishes owing to the overwhelming $\lambda$ and the associated positive keys, and JCL has no motivation to distinguish between positive and negative keys. From Fig.(2(b)), we can see that the performance is relatively stable in a wide range of [0.2,4.0]. **3)** $\tau$: The temperature $\tau$ [22] affects the flatness of softmax function and the confidence of each positive pair. From Fig.(2(c)), the optimal $\tau$ turns out to be around 0.2. As $\tau$ increases beyond 0.2, the classification accuracy starts to drop, owing to an increasing uncertainty and reduced confidence of the positive pairs. When the value $\tau$ becomes too small, the algorithm tends to overweight the influence of each positive pair and degrade the pre-training.

## 4.5 Feature Distribution

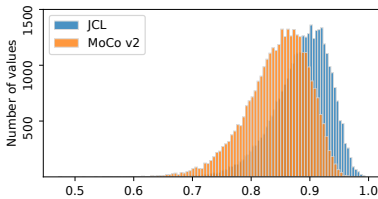

(a) Similarity distribution

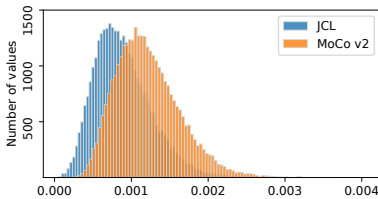

(b) Variance distribution

Figure 3: Distribution of positive pair similarities and feature variances.

In section 3.3, we hypothesis that JCL favors consistent features across distinct $\boldsymbol{k}_i^+$ owing to the application of Jensen inequality, and therefore would force the network to find invariant representation across different positive keys. These invariant features are the core mechanism that makes JCL a good pre-training candidate for obtaining good generalization capabilities. To validate this hypothesis, we qualitatively measure and visualize the similarities of positive samples within each positive pair. To be more concrete, we randomly sample 32,768 images from ImageNet100, and generate 32 different augmentations for each image (see supplementary material for more detailed settings). We feed these images respectively into the JCL pre-trained and MoCo v2 pre-trained ResNet-18 network and then directly extract the features out of each ResNet-18 network. Firstly, we use these features for calculating the cosine similarities of each pair of features (every pair of 2 out of 32 augmentations) belonging to the same identity image. In other words, $32 \times 32$ cosine similarities for each image are averaged into a single sample point (therefore, 32,768 points in total). Fig.(3(a)) illustrates the histogram of these cosine similarities. It is clear that JCL achieves much more samples with higher similarity scores than that of the MoCo v2. This implies that JCL indeed tends to favor a more consistent feature invariance

within each instance-specific distribution. We also extract the diagonal entries from $\Sigma_{k_i^+}$ and display the histogram in Fig.(3(b)). Accordingly, the variance of the obtained features belonging to the same image is much smaller, as shown in Fig.(3(b)). This also aligns with our hypothesis that JCL favors consistent representations across different positive keys.

## 5  Conclusions

We propose a particular form of contrastive loss named joint contrastive learning (JCL). JCL implicitly involves the joint learning of an infinite number of query-key pairs for each instance. By applying rigorous bounding techniques on the proposed formulation, we transfer the originally intractable loss function into practical implementations. We empirically demonstrate the correctness of our bounding technique along with the superiority of JCL on various benchmarks. These empirical evidences also qualitatively support our theoretical hypothesis behind the central mechanism of JCL. Most notably, although JCL is an unsupervised algorithm, the JCL pre-trained networks even outperform its supervised counterparts in many scenarios.

## Broader Impact

Supervised learning has seen tremendous success in the AI community. By heavily relying on human annotations, supervised learning allows for convenient end-to-end training of deep neural networks. However, label acquisition is usually time-consuming and economically expensive. Particularly, when the algorithm needs to pre-train on massive datasets such as ImageNet, obtaining the labels for millions of data becomes an extremely tedious and expensive prerequisite that hinders one from trying out interesting ideas. This significantly limits and discourages the motivations for relatively small research communities without adequate financial supports. Another concern is the accuracy of the annotations, as labeling millions of data might very likely induce noisy and wrong labels owing to mistakes. What we have proposed in this paper is an unsupervised algorithm called JCL that solely depends on data itself without human annotations. JCL offers an alternative way to more efficiently exploit the pre-training dataset in an unsupervised way. One can even build up his/her own pre-training dataset by crawling data randomly from internet without any labeling efforts. However, one potential risk lies in the fact that if the usage of unsupervised visual representation learning aims at visual understanding systems (e.g., image classification and object detection), these systems may now be easily approached by those with lower levels of domain knowledge or machine learning expertise. This could expose the visual understanding model to some inappropriate usage and occasions without proper regulation or expertise.

## Acknowledgments and Disclosure of Funding

Funding in direct support of this work: Financial support from JD AI research. There is no additional source of revenue paid or to be paid related to this work.

## Footnotes

*Qi Cai and Yu Wang contributed equally to this work. This work was performed at JD AI Research.

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
