[Supplementary Material]

# Joint Contrastive Learning with Infinite Possibilities — Supplementary Materials

**Qi Cai**[1]*   **Yu Wang**[2]*   **Yingwei Pan**[2]   **Ting Yao**[2]   **Tao Mei**[2]
[1] University of Science and Technology of China, Hefei, China
[2] JD AI Research, Beijing, China
{cqcaiqi, feather1014, panyw.ustc, tingyao.ustc}@gmail.com, tmei@live.com

The supplementary material contains: **1)** the theoretical derivation of Eq.(M.8); **2)** the computational complexity analysis of JCL; **3)** the implementation details of pre-training on ImageNet1K ( as in Section (M.4.1)); **4)** the experimental settings of the ablation studies (as in Section (M.4.4)); **5)** the details for visualizing similarities and variances distributions (as in Fig.(M.3)).

*Note: We use notation Eq.(M.xx) to refer to the equation Eq.(xx) presented in the main paper, and use Eq.(S.xx) to indicate the equation Eq.(xx) in this supplementary material. Similarly, we use Fig./Section/Table (M.xx) and Fig./Section/Table (S.xx) to respectively indicate a certain figure/section/table in the main paper (M.xx) or supplementary material (S.xx).*

## 1   Theoretical Derivation of Eq.(M.8)

Firstly, for any random variable $x$ that follows Gaussian distribution $x \sim \mathcal{N}(\mu, \Sigma)$, where $\mu$ is the expectation of $x$, $\Sigma$ is the variance of $x$, we have the moment generation function that satisfies:

$$\mathbb{E}_x[e^{a^T x}] = e^{a^T \mu + \frac{1}{2} a^T \Sigma a}. \tag{1}$$

Under the Gaussian assumption $k_i^+ \sim \mathcal{N}(\mu_{k_i^+}, \Sigma_{k_i^+})$, along with Eq.(S.1), we find that Eq.(M.7) immediately reduces to:

$$\text{Eq.(M.7)} = \log\left[\exp(q_i^T \mu_{k_i^+}/\tau + \frac{1}{2\tau^2} q_i^T \Sigma_{k_i^+} q_i) + \sum_{j=1}^{K} \exp(q_i^T k_{i,j}^-/\tau)\right] - q_i^T \mu_{k_i^+}/\tau. \tag{2}$$

Since the statistics are more/less informative in the later/beginning of the training, we scale the influence of $\Sigma_{k^+}$ by multiplying it with a scalar $\lambda$. Such tuning of $\lambda$ hopefully stabilizes the training, leading to a modified version of Eq.(S.2):

$$\bar{\mathcal{L}}_i = \log\left[\exp(q_i^T \mu_{k_i^+}/\tau + \frac{\lambda}{2\tau^2} q_i^T \Sigma_{k_i^+} q_i) + \sum_{j=1}^{K} \exp(q_i^T k_{i,j}^-/\tau)\right] - q_i^T \mu_{k_i^+}/\tau. \tag{3}$$

This resembles our derivation of Eq.(M.8) in the main paper.

## 2   Computational complexity

Regarding the GPU memory cost, the vanilla formulation explicitly involves a batchsize that is $M$ times larger than the conventional contrastive learning. In contrast, the batchsize for JCL remains the same as the conventional contrastive learning. Although $M'$ positive keys are required to compute the sufficient statistics for JCL, the batchsize and the incurred multiplications (between query and key) are reduced. Particularly, JCL utilizes multiple keys merely to reflect the statistics, which helps JCL more efficiently exploit these samples. Therefore, JCL is more memory efficient than vanilla when they achieve the same performance, and JCL always offers better performance when the both cost similar memories.

# 3    Implementation Details of Pre-Training on ImageNet1K

For all the experiments, we generate augmentations in the same way as in MoCo v2 [1] for pre-training. First, a 224×224 patch is randomly cropped from the resized images. Then color jittering, random grayscale, Gaussian blur, and random horizontal flip are sequentially applied to each patch. We implement ShuffleBN in [2] by concatenating $N \times M'$ positive keys in the batch dimension and shuffling the data before feeding them into the network. In regard of negative samples presented in Eq.(M.8), we update the queue $\mathcal{Q}$ following the design in [2]. In detail, we enqueue the average values of keys $\{\boldsymbol{\mu}_{k_i^+}\}_{i=1}^N$ during training and dequeue the oldest (256 number of) keys in $\mathcal{Q}$. The momentum value [2] for updating the key encoder is 0.999 and the queue size is 65,536. For pre-training, we use SGD with 0.9 momentum and 0.0001 weight decay. The pre-trained weights of query encoder are extracted as network initialization for downstream tasks.

# 4    Experimental Settings of Ablation Studies

The ablation experiments are conducted on a subset of ImageNet1K (i.e., ImageNet100) following [4]. Specifically, 100 classes are randomly sampled from the primary ImageNet1K dataset, which are utilized for both pre-training and linear classification. The exact classes include:

n02869837, n01749939, n02488291, n02107142, n13037406, n02091831, n04517823, n04589890, n03062245, n01773797, n01735189, n07831146, n07753275, n03085013, n04485082, n02105505, n01983481, n02788148, n03530642, n04435653, n02086910, n02859443, n13040303, n03594734, n02085620, n02099849, n01558993, n04493381, n02109047, n04111531, n02877765, n04429376, n02009229, n01978455, n02106550, n01820546, n01692333, n07714571, n02974003, n02114855, n03785016, n03764736, n03775546, n02087046, n07836838, n04099969, n04592741, n03891251, n02701002, n03379051, n02259212, n07715103, n03947888, n04026417, n02326432, n03637318, n01980166, n02113799, n02086240, n03903868, n02483362, n04127249, n02089973, n03017168, n02093428, n02804414, n02396427, n04418357, n02172182, n01729322, n02113978, n03787032, n02089867, n02119022, n03777754, n04238763, n02231487, n03032252, n02138441, n02104029, n03837869, n03494278, n04136333, n03794056, n03492542, n02018207, n04067472, n03930630, n03584829, n02123045, n04229816, n02100583, n03642806, n04336792, n03259280, n02116738, n02108089, n03424325, n01855672, n02090622

For ablation studies, we adopt ResNet-18 as the backbone and all models are trained for 100 epochs with a batch size of $N = 128$ at the pre-training stage. The learning rate is set to $lr = 0.1$ and is gradually annealed following a cosine decay schedule [3]. For linear classification, all models are trained for 100 epochs with a learning rate of $lr = 10.0$. The learning rate is decreased by 0.1 at $60^{th}$ and $80^{th}$ epochs, and the batch size is $N = 256$. Table 1 shows the detailed settings of hyper-parameters for three ablation experiments of the main paper.

Table 1: Hyper-parameters for ablation studies in Section (M.4.4) of the main paper. " / " indicates non-applicable, since the corresponding hyper-parameter is varied for ablation study.

| Ablation Studies | Number of positive keys $M'$ | Strength of augmentation $\lambda$ | Temperature for softmax $\tau$ |
|---|---|---|---|
| Fig.(M.2(a)) | / | 4.0 | 0.2 |
| Fig.(M.2(b)) | 5 | / | 0.2 |
| Fig.(M.2(c)) | 5 | 4.0 | / |

# 5    Details for Visualizing Similarities and Variances Distributions

For the experiments that visualize the distributions of similarities and variances in Section (M.4.5), we respectively extract features from pre-trained ResNet-18 models of JCL and MoCo v2. At the pre-training stage, the hyper-parameters are set as $\tau = 0.2$, $M' = 9$ and $\lambda = 4.0$ for JCL. For MoCo v2, we use the released code from https://github.com/facebookresearch/moco to train a ResNet-18 model. Both JCL and MoCo v2 are trained on the ImageNet100 dataset for 100 epochs. Other hyper-parameters are exactly the same as the settings used in ablation studies. In total, we sample 32,768 images for depicting the histograms in Fig.(M.3). For each image, we randomly generate 32 augmented images and feed these images into the pre-trained network to extract features. The feature vectors are $\ell_2$ normalized before computing similarities and variances. For the similarity visualization in Fig.(M.3(a)), cosine similarities of all pairs ($32 \times 32$ in total) are averaged into a single sample point used for drawing the histogram (hence 32,768 points in Fig.(M.3(a))). Similarly, for visualizing feature variances Fig.(M.3(b)), we use the $\ell_2$ normalized features to compute the

covariance matrix of augmented images belonging to the same source image, and we average the diagonal values of covariance matrix for each source image into a single sample point to draw the histogram (hence 32,768 points in Fig.(M.3(b))). Note that only using diagonal values of the covariance matrix respects our primary purpose of computing the $dimension - dimension$ feature correlations between feature vectors.

## Footnotes

*Qi Cai and Yu Wang contributed equally to this work. This work was performed at JD AI Research.