[Reviews · NeurIPS 2020]

Review 1

Summary and Contributions: In this paper, the authors focus on unsupervised representation learning. To this end, they propose a variation of contrastive learning: joint constrastive learning with infinitely many keys. The authors justify their approach mathematically and show competitive performance against baselines such as MoCo. They also perform a thorough ablation analysis and demonstrate the invariance of the learned features.

Strengths: 1. The paper address a timely topic in the machine learning and computer vision community. Unsupervised learning can allow our models to be far more data efficient. 2. The authors strongly ground their approach mathematically. They explicitly justify their loss function through mathematical derivation. 3. Performance against baselines is competitive and convincing. The ablation analysis is thorough.

Weaknesses: 1. The technical novelty is limited. JCL is yet another variation of contrastive learning approaches to representation learning. The main novel idea is the approximate use of a large number of keys to better enforce feature invariation. 2. While this is an issue with many contrastive papers in general, the authors don't evaluate their approach beyond the domain of computer vision. While the approach should apply to data in general, it is unknown in practice how well it would generalize to data modalites such as sound, language, video, etc.

Correctness: Yes. The methodology appears to be correct.

Clarity: Yes, the paper is very clear.

Relation to Prior Work: Yes, the authors evaluated their approach against prior work such as MoCo.

Reproducibility: Yes

Additional Feedback: After Rebuttal: I have read the rebuttal. The authors have addressed my concerns. I keep my rating.


Review 2

Summary and Contributions: This paper considers the question of how to modify the typical contrastive objective so as to encourage invariance across multiple augmented inputs simultaneously. They provide a simple method based on considering points from the same similarity group to be encoded as samples from a Gaussian key value, and taking expectations over this key value. The main technical tool was an application of Jensen's inequality to yield an objective for which empirical versions of the mean and covariance matrix can be plugged in. They evaluate their method on a variety of downstream image understanding tasks, such as ImageNet classification and MS COCO segmentation & keypoint detection, showing competitive performance compared to baselines and contemporary methods. They also included ablation studies comparing and contrasting how their method behaves as you vary various hyperparameters (finding that some of the behavior is qualitatively different from other methods, particularly Fig2(b)). Overall I found the results to be interesting, and feel that, combined with the ablation studies, they constitute a sufficiently detailed exploration of the topic.

Strengths: - Method is simple (I view this as a strength) and builds on a natural conjecture one would have about contrastive learning (that more positive samples should help). Because of this I find the results quite believable. - Method beats state of the art baselines in various image understanding tasks. - Ablation studies were interesting and illustrative. Particularly Fig3 yields good evidence as to why the method is improving performance.

Weaknesses: - Bad quality of written language (see details below). - Some experimental details aren't clear (see "correctness" section for details).

Correctness: The method seems broadly believable. I have several questions about the experiments I would be interested in getting input from the authors on: - I did not fully follow why the comment in l.179-183 is true, could you clarify? Specifically, I see why the vanilla method increases memory requirements by a factor of M', but am unclear on the case of your method. Is it because of the sufficient statistic property? - I would be curious to know how well the vanilla baseline does. It would be good to clarify to the reader which of the following two cases we are in: 1) your method both improves downstream performance AND memory cost, or 2) your method is exchanging some (perhaps slight) decrease in accuracy IN EXCHANGE for reduced memory cost. - Can you clarify where the competitor results in Table 2 for SimCLR come from? A search of the original SimCLR paper shows that the number “87.3” doesn’t appear anywhere, yet you quote it as the top-5 accuracy. This makes me a little confused. If you have reimplemented SimCLR it would be good to make this clear (apologies if you did and I missed that). - You say that you use the same hyperparameters as MoCo v2 which takes \tau=0.07, reporting a JCL accuracy of 68.7% in Fig1 for ImageNet. But in Fig2(c) \tau=0.07 is reported as giving an accuracy ~66%. What am I missing here? Did you also take \tau=0.07 for SimCLR? (Perhaps that explains the previous question, since in SimCLR they take \tau=0.5.) - Why does Table 1 include MoCo v2 as a baseline but Table 2 doesn't? Since MoCo v2 easily beats MoCo in Table 1 and JCL sometimes only narrowly beats MoCo in Table 2 this makes me wonder if MoCo v2 could beat JCL in Table 2. Clarification of this question is important for assessing the value of Table 2. - Are the results in Fig3(b) obtained by computing the variance of the set of real numbers diag(\Sigma)?

Clarity: To put it briefly, the quality of writing in this paper in its present state is simply not up to scratch. I do not mean to be too discouraging, since writing is something that can be refined and improved, and shouldn’t be a fatal judgement on a paper. However the problems in the written language in this paper are multifaceted and affect various aspects of the work. Before publication at NeurIPS - or any other venue - a significant amount of polishing is in order. Please see the additional feedback section for details (some of which are merely personal stylistic points, others more directly affect the message of the paper).

Relation to Prior Work: A related works & comparison to existing works sections are included. The contribution of the paper is clearly distinct from the mentioned literature, and from other works I am aware of in the area. Broadly there are no glaring emission in terms of citations. While certainly not essential, the authors could consider mentioning related work in other data modalities such as NLP and RL, for example, - "An efficient framework for learning sentence representations." https://arxiv.org/abs/1803.02893 -"CURL: Contrastive Unsupervised Representations for Reinforcement Learning" https://proceedings.icml.cc/static/paper_files/icml/2020/5951-Paper.pdf For your interest, there is a recent work which also (in part) introduces a method for using multiple positive samples at once in a memory efficient way, and also shows this improved the learned representation. "Debiased Contrastive Learning" https://arxiv.org/pdf/2007.00224.pdf (no need for you to compare your work to it since this one is so recent, I just mention it for your interest)

Reproducibility: Yes

Additional Feedback: I think it is too strong to claim that “we also theoretically unveil the certain important mechanisms that govern the behavior of JCL.” The main theoretical tool in the proposed method is an application of Jensen’s inequality. There is also a section (3.3) that discusses some very basic properties of the the objective. To claim any of this as a significant “theoretical contribution” is too strong in my view. To me, the most interesting aspect of Fig2 is part (b). In my mind the reason this differs from the observations of other contrastive methods is as follows: data augmentations used in contrastive methods preserve semantic content. On the other hand, the \lambda parameter acts as a scaling on the covariance matrix which, when made too large, causes semantically different inputs' distributions to overlap in latent space, resulting in non-identifiability. Do the authors have additional insight into why this is the case? Some discussion on this point could be valuable in the paper since it is a case where the characteristics of your method differ from the norms observed in contemporary work. The authors are quite defensive over their paper's relation to CMC. While I personally think this work is sufficiently distinct from CMC, and the underlying reasons given are fair, I would kindly encourage the authors to rephrase this section. There is not need to be so defensive in my opinion. - I think there is a typo on Algorithm 1 line 8. It should be g(x)'s instead of just x's. Aspects of the writing that need work: (examples are just examples to illustrate the point) -- overly colorful language. Examples: l.2 “novel probabilistic attire”, l.8 "also theoretically unveil the certain important mechanisms”, l.9 “We demonstrate that the proposed formulation harbors an innate agency”, l.22 “This overcomes several perils that the supervised learning encounters”, l.44 “This allows us to introduce a probabilistic tool and thus absorb the impact of large number of positive pairs via rigorous bounding techniques.”, l.45 “We could handily approach an analytic form of gradients for backpropagation”, l.77 “However, both our bounding technique and the motivation of how “infinity” is involved”, l.162 “This is where the data statistics come to rescue.”, l.201 “we empirically evaluate and analyze the hypotheses that directly emanated from the design of JCL”. -- nonsensical words. Examples: l.113 “unsupervisely”, l.286 “Reversely” -- Loose phrasing that causes claims to be vague, and sometimes appear to be over-claims. Examples: l.77 “The concept of infinity has also been shown advantageous in ISDA” l.157 “where Eq.(7) tightly upperbounds” I think it is an overclaim to say that applying Jensen’s inequality is a tight upper bound. I would take more care over the claim and qualify that it is tight in the best case where k_i^+ is constant. l.8 “we also theoretically unveil the certain important mechanisms that govern the behavior of JCL.” I assume this refers to section (3.3) that discusses some very basic properties of the the objective. I understand what you mean, but would avoid seeming like you are claiming a “theoretical contribution” here. Update: conditional on the authors adding in the changes they mentioned in their rebuttal, I am happy to accept their comments and confirm my review as an accept.


Review 3

Summary and Contributions: The paper focused on using a parametric model to characterize the distribution of multiple positive keys, which is an interesting step beyond the current standard one-query-one-positive-key setting. The idea is to assume a Gaussian distribution of the final embedding vectors from the same instance-specific class, with the statistics estimated on-line using multiple crops of the same image for positive keys (instead of just one). The network is shown to improve over the strong baseline established by MoCo v2. Detailed analysis and visualizations are provided.

Strengths: + I think this is a good paper that puts forward a simple core idea, shows it being helpful, and gives analysis/insights that justify how it works. Most of the statements/claims in this paper are sound. + Overall it is an interesting idea that explores how to efficiently use multiple crops for the current contrastive learning. It could be of interest to the community.

Weaknesses: - My biggest concern of the paper is the potential diminishing improvement as one trains longer. MoCo v2 has been shown to benefit a lot from longer training, which could be explained by seeing more examples allows it build better and better instance discriminators for each image. While I believe it is indeed beneficial for the network to model the set of positive crops in the shorter learning schedule (200 ep), such gains may be diminishing as one goes on to train longer and longer. I am happy to be convinced otherwise in the rebuttal.

Correctness: While I haven't checked all the equations in 100% details, I am 99% sure the derivations and technique is sound and correct.

Clarity: This is a very well written paper. The approach motivates the problem, the naive solution, and leads to the proposed solution in a very organized, story-telling way. The experiments are also organized well, with both final results and detailed analysis.

Relation to Prior Work: The work is building on state-of-the-art methods. The idea is sufficiently new to me.

Reproducibility: Yes

Additional Feedback:


Review 4

Summary and Contributions: This paper proposes a novel loss for self-supervised contrastive representation learning methods. Current contrastive learning methods mostly rely on generating the query-key pairs for self-supervised training. Most of them penalize the single positive query-key pairs independent of other positive query-key pairs of a single training sample in their loss functions. This way of independent penalization, therefore, results in neglecting the assumption that all augmentations corresponding to a specific training sample are statistically dependent on each other. The authors use this as their stepping stone toward their novel loss. They aim to introduce a novel loss that encourages the similarity consistency withing each instance-specific class. Therefore, they consider simultaneously penalizing multiple positive query-key pairs by their in-pair dissimilarity. To achieve such a goal, they need to increase the number of positive query-key pairs. Having multiple positive pairs, the loss would be the average sum of the InfoNCE loss. However, this action will cost in higher memory usage and will be capped by the hardware memory capacity. To alleviate this limitation, they propose to push the limit to infinity. Mathematically, the estimate of the averaged sum of InfoNCE losses for infinite pairs would be Eq.(4). By applying the Jensen Inequality, authors estimate the upper bound for the loss in Eq.(7). Assuming that the set of positive keys follows Gaussian distribution, the final loss would be Eq.(8). Experiments on different downstream tasks such as linear classification on ImageNet1K, object detection, instance segmentation, and keypoint detection on MS COCO show the superiority of the proposed method.

Strengths: + Proposing a novel loss that not only benefits the shared similarity between the augmentation in contrastive learning methods but also resolves the memory limitation issues by estimating the infinity upper bound + Novel solution for estimating the upper bound of the loss + The contributions show significant improvement over a wide variety of downstream tasks

Weaknesses: - The second contribution mentioned in the introduction, is not well addressed in the rest of the paper, especially in the experiments. - Although there are plenty of details provided regarding the implementation details and hyperparameter settings, authors have not included the code and there's no commitment regarding publishing the code. - In table 1, it is not clear why some methods are trained for 200 epochs.

Correctness: The claims regarding the limitations of the current existing methods are correct. The experiments support most of the claims about the proposed method. However, some claims mentioned in the contributions are not clearly discussed and supported in the experiments. Look at the "Weaknesses" section.

Clarity: The paper is well written. The notations are nicely described. However, there are a few points that could be improved: - The second contribution mentioned in the introduction: sentencing is too general and a little bit unclear. - It's better to move most of the sub-section 3.1 to the related work and discuss the current methods there. - The abstract is not fully representing the contributions and novelty of the paper

Relation to Prior Work: The authors have clearly explained the differences to the previous works in sections 2 and 3 (3.1).

Reproducibility: Yes

Additional Feedback: I have read the author's response and my final score would be accept.

[Author Response · NeurIPS 2020]

#**Reviewer1**. Thank you for acknowledging the key contributions of our paper. **R1.1 Novelty:** The novel practical impacts of our work are as follows. Firstly, to our best knowledge, JCL is the first that attempts to address the contrastive learning problem where infinite many of keys are considered. Secondly, by capitalizing on the proposed probabilistic model, JCL eases the aforementioned infinity scenario by simply using only a few key samples. Thirdly, JCL relies on a memory economical calculation, while its vanilla multi-key counterpart is less memory efficient when achieving similar performance (also see **R2.1 & R2.2**). We believe that all of these nontrivial efforts explore and shed further light on important properties of contrastive learning. **R1.2 Generalize to video:** As suggested, we conducted additional experiments on video data (UCF101) to evaluate the JCL pre-trained features for action recognition downstream task via linear classification. The top-1 accuracy of JCL pre-trained features is 48.6%, which outperforms MoCo v2 (47.3%). Generalization of JCL for other data modalities (sound, language, video) will be included in our future work.

#**Reviewer2**. We appreciate the comments. Regarding your concerns of the written quality and typos (e.g., Algorithm 1 line 8), we would carefully polish throughout the paper. **R2.1 & R2.2 Memory and accuracy:** Regarding the memory cost, the vanilla explicitly involves a batchsize that is $M$ times larger than the conventional contrastive learning. In contrast, the batchsize for JCL remains the same as the conventional contrastive learning. Although $M'$ positive keys are required to compute the sufficient statistics for JCL, the batchsize and the incurred multiplications (between query and key) are reduced. Particularly, JCL utilizes multiple keys merely to reflect the statistics, which helps JCL more efficiently exploit these samples. Therefore, JCL is more memory efficient than vanilla when they achieve the same performance, and JCL always offers better performance when the both cost similar memories. For fair comparison, we run the additional experiments to compare vanilla against JCL when the number of positive keys used is identical (i.e., $M = M' = 5$). The top-1 accuracy on ImageNet100 for vanilla (ResNet-50) is 80.9% while JCL achieves 82.0%. We will add the discussions. **R2.3 SimCLR:** The top-5 accuracy we reported (87.3%) for SimCLR was extracted from the primary Fig. B.1 in the arxiv version of SimCLR. **R2.4 Hyperparameters:** The experiments in Table1 are performed on ImageNet1K with ResNet-50, whereas all the experiments for Figure2 correspond to the training on a *subset* of ImageNet1K (ImageNet100 [37]) with ResNet-18. The detailed setting of Figure2 can be referred to Supplementary Material (Section S.3). Thus, there is no one-one correspondence between the data in Table1 and Figure2. In fact, MoCo v2 [8] uses $\tau = 0.2$, which differs from $\tau = 0.07$ used in MoCo [17]. For SimCLR, we directly extracted the results from the paper, which uses default $\tau = 0.1$ ($\tau = 0.5$ is used for CIFAR10 instead of ImageNet). **R2.5 MoCo v2 in Table 2:** The MoCo v2 paper [8] only reports the results on ImageNet1K and VOC (a relatively small dataset for object detection), while our paper instead presents the results on MS COCO (more challenging benchmark for object detection & instance segmentation). As suggested, we conducted additional experiments and evaluated MoCo v2 on MS COCO for object detection & instance segmentation tasks. The results are 37.6% ($AP^{bb}$) and 35.3% ($AP^{mk}$), which are lower than ours (38.1% $AP^{bb}$, 35.6% $AP^{mk}$). We will add the results. **R2.6 Fig.3(b):** Yes, we use $diag(\Sigma_{k_i^+})$ for the histogram in Fig.3(b). **R2.7 Impact of $\lambda$:** According to Gaussian model and Eq.(8), a relatively small $\lambda$ introduces reasonable variance into the mean positive keys that is beneficial for JCL, as more variants of positive key are potentially included. But when $\lambda$ grows large enough, the introduced variance $\Sigma_{k_i^+}$ starts to ***dominate*** the positive mean $\boldsymbol{\mu}_{k_i^+}$ value. This large $\lambda$ therefore will distort the magnitude scale of positive keys and dilute the impact of negative keys in Eq.(8). Consequently, the distribution of the $\boldsymbol{k}_{i,m}^+$ and $\boldsymbol{k}_{i,j}^-$ would also significantly be distorted. When $\lambda$ grows to infinity, the effect of negative keys completely vanishes owing to the overwhelming $\lambda$ and the associated positive keys, and JCL has no motivation to distinguish between positive and negative keys. **R2.8 References:** Thanks for the references, we would carefully check these and discuss the differences in detail.

#**Reviewer3**. Many thanks for the positive comments. #**R3.1 More iterations:** As MoCo sees more samples, JCL also benefits from seeing more variants of positive keys (owing to the role of $\Sigma_{k_i^+}$, see Algorithm 1) as training proceeds longer. In addition, JCL enforces intra-class invariance, and thus intrinsically offers structural

Figure 1: Accuracy on ImageNet100.

| Epochs | 200 | 300 | 400 | 500 |
|---|---|---|---|---|
| MoCo v2 | 80.0 | 82.7 | 83.5 | 84.1 |
| JCL | 82.0 | 83.9 | 84.5 | 85.0 |

advantage that is absent in MoCo, even if both run till the convergence. To verify this, we run JCL and MoCo v2 for more epochs ([200, 500]) and evaluated both of the algorithms on linear classification in a subset of ImageNet1K (ImageNet100 [37]). As shown in Figure 1, although the performance gap between the two indeed drops from 2.0% to 1.2% when the epoch number increases to 300, we observe consistent performance gains ($\sim 1\%$) offered by JCL when epoch number $> 300$. The results further justify the effectiveness of JCL even in longer iterations.

#**Reviewer4**. **R4.1 Second contribution:** Thanks. Regarding the second contribution, Line 46 means the derivations in our algorithm lead to analytic closed form of gradients that can be easily approached, e.g., by using PyTorch (although we did not explicitly present the calculation of these gradients and these could be added in the revision). Lines 47-48 indicate that the experiments in Fig. 3 justify the hypotheses in Sec. 3.3 (e.g., JCL favors a more consistent feature invariance within each instance-specific distribution). We will rephrase these to ensure clarity. **R4.2 Code:** We actually provided the link of all source codes in supplementary material. **R4.3 200 epochs:** For fair comparisons with the most recent competitors (SimCLR, MoCo, MoCo v2) under the same epochs, we follow the commonly adopted settings (200 epochs) as in [8] to pre-train our JCL. **R4.4 Writing:** Thanks, we will carefully modify the paper structure.

[Meta-Review · NeurIPS 2020]

This paper achieved a high accept consensus. The paper puts forward a simple core idea, shows it being helpful, and gives analysis/insights that justify how it works. The authors strongly ground their approach theoretically. The method beats SOTA in various tasks. However, a bad quality of language and some experimental details missing were reported and I encourage the authors to fix these following the reviewer's recommendations for the final version of the manuscript.